# THE ADVANTAGE REGRET-MATCHING ACTOR-CRITIC

## ABSTRACT

Regret minimization has played a key role in online learning, equilibrium computation in games, and reinforcement learning (RL). In this paper, we describe a general model-free RL method for no-regret learning based on repeated reconsideration of past behavior: Advantage Regret-Matching Actor-Critic (ARMAC). Rather than saving past state-action data, AR-MAC saves a buffer of *past policies*, replaying through them to reconstruct hindsight assessments of past behavior. These retrospective value estimates are used to predict conditional advantages which, combined with regret matching, produces a new policy. In particular, ARMAC learns from sampled trajectories in a centralized training setting, without requiring the application of importance sampling commonly used in Monte Carlo counterfactual regret (CFR) minimization; hence, it does not suffer from excessive variance in large environments. In the single-agent setting, ARMAC shows an interesting form of exploration by keeping past policies intact. In the multiagent setting, ARMAC in self-play approaches Nash equilibria on some partially-observable zero-sum benchmarks. We provide exploitability estimates in the significantly larger game of betting-abstracted no-limit Texas Hold'em.

## 1 INTRODUCTION

The notion of regret is a key concept in the design of many decision-making algorithms. Regret minimization drives most bandit algorithms, is often used as a metric for performance of reinforcement learning (RL) algorithms, and for learning in games (3). When used in algorithm design, the common application is to accumulate values and/or regrets and derive new policies based on these accumulated values. One particular approach, counterfactual regret (CFR) minimization (35), has been the core algorithm behind super-human play in Computer Poker research (4; 25; 6; 8). CFR computes an approximate Nash equilibrium by having players minimize regret in self-play, producing an average strategy that is guaranteed to converge to an optimal solution in two-player zero-sum games and single-agent games.

We investigate the problem of generalizing these regret minimization algorithms over large state spaces in the sequential setting using end-to-end function approximators, such as deep networks. There have been several approaches that try to predict the regret, or otherwise, simulate the regret minimization: Regression CFR (RCFR) (34), advantage regret minimization (17), regret-based policy gradients (30), Deep Counterfactual Regret minimization (5), and Double Neural CFR (22). All of these approaches have focused either on the multiagent or single-agent problem exclusively, some have used expert features, while others tree search to scale. Another common approach is based on fictitious play (15; 16; 21; 24), a simple iterative self-play algorithm based on best response. A common technique is to use reservoir sampling to maintain a buffer that represents a uniform sample over past data, which is used to train a classifier representing the average policy. In Neural Fictitious Self-Play (NFSP), this produced competitive policies in limit Texas Hold'em (16), and in Deep CFR this method was shown to approach an approximate equilibrium in a large subgame of Hold'em poker. A generalization of fictitious play, policy-space response oracles (PSRO) (21), stores past policies and a meta-distribution over them, replaying policies against other policies, incrementally adding new best responses to the set, which can be

seen as a population-based learning approach where the individuals are the policies and the distribution is modified based on fitness. This approach only requires simulation of the policies and aggregating data; as a result, it was able to scale to a very large real-time strategy game (33). In this paper, we describe an approximate form of CFR in a training regime that we call *retrospective policy improvement*. Similar to PSRO, our method stores past policies. However, it does not store meta-distributions or reward tables, nor do the policies have to be approximate best responses, which can be costly to compute or learn. Instead, the policies are snapshots of those used in the past, which are retrospectively replayed to predict a conditional advantage, which used in a regret matching algorithm produces the same policy as CFR would do. In the single-agent setting, ARMAC is related to POLITEX (1), except that it is based on regret-matching (14) and it predicts average quantities rather than explicitly summing over all the experts to obtain the policy. In the multiagent setting, it is a sample-based, model-free variant of RCFR with one important property: it uses trajectory samples to estimate quantities *without requiring importance sampling* as in standard Monte Carlo CFR (20), hence it does not suffer from excessive variance in large environments. This is achieved by using critics (value estimates) of past policies that are trained off-policy using standard policy evaluation techniques. In particular, we introduce a novel training regime that estimates a conditional advantage $W_i(s, a)$, which is the cumulative counterfactual regret $R_i(s, a)$, scaled by factor $B(s)$ that depends on the information state $s$ only; hence, using regret-matching over this quantity yields the policy that CFR would compute when applying regret-matching to the same (unscaled) regret values. By doing this entirely from sampled trajectories, the algorithm is model-free and can be done using any black-box simulator of the environment; hence, ARMAC inherits the scaling potential of PSRO without requiring a best-response training regime, driven instead by regret minimization.

**Problem Statement**. CFR is a tabular algorithm that enumerates the entire state space, and has scaled to large games through domain-specific (hand-crafted) state space reductions. The problem is to define a model-free variant of CFR using only sampled trajectories and general (domain-independent) generalization via functional approximation, *without* the use of importance sampling commonly used in Monte Carlo CFR, as it can cause excessive variance in large domains.

## 2 BACKGROUND

In this section, we describe the necessary terminology. Since we want to include the (partially-observable) multiagent case and we build on algorithms from regret minimization we use extensive-form games notations (29). A single-player game represents the single-agent case where histories are aggregated appropriately based on the Markov property.

A **game** is a tuple $(\mathcal{N}, \mathcal{A}, \mathcal{S}, \mathcal{H}, \mathcal{Z}, u, \tau)$, where $\mathcal{N} = \{1, 2, \cdots, n\}$ is the set of players. By convention we use $i \in \mathcal{N}$ to refer to a player, and $-i$ for the other players ($\mathcal{N} - \{i\}$). There is a special player $c$ called *chance* (or *nature*) that plays with a fixed stochastic strategy (chance's fixed strategy determines the transition function). $\mathcal{A}$ is a finite set of actions. Every game starts in an initial state, and players sequentially take actions leading to histories of actions $h \in \mathcal{H}$. Terminal histories, $z \in \mathcal{Z} \subset \mathcal{H}$, are those which end the episode. The utility function $u_i(z)$ denotes the player $i's$ return over episode $z$. The set of states $\mathcal{S}$ is a partition of $\mathcal{H}$ where histories are grouped into **information states** $s = \{h, h', \ldots\}$ such that the player to play at $s$, $\tau(s)$, cannot distinguish among the possible histories (world states) due to private information only known by other players [1]. Let $\Delta(X)$ represent all distributions over $X$: each player's (agent's) goal is to learn a policy $\pi_i : \mathcal{S}_i \to \Delta(\mathcal{A})$, where $\mathcal{S}_i = \{s \mid s \in \mathcal{S}, \tau(s) = i\}$. For some state $s$, we denote $\mathcal{A}(s) \subseteq \mathcal{A}$ as the legal actions at state $s$, and all valid state policies $\pi(s)$ assign probability 0 to illegal actions $a \notin \mathcal{A}(s)$.

We now show a diagram to illustrate the key ideas. Kuhn poker, shown in Figure 1 is a poker game with a 3-card deck: Jack (J), Queen (Q), and King (K). Each player antes a single chip and has one more chip to bet with, then gets a single priavte card at random and one is left face down, and players proceed to bet (b) or pass (p). Initially the game

---

[1]Information state is the belief about the world that a given player can infer based on her limited observations and may correspond to many possible histories (world states)

starts in the empty history $h_0 = \emptyset$ where no actions have been taken, and it is chance's turn to play. Suppose chance samples, according to a fixed distribution, one of its six actions, which corresponding to one of the size-2 permutations of deals (one card to each player). For example, suppose outcome `1Q2J` is sampled, corresponding to the first player getting the queen and second player getting the jack. This would correspond to a new history $h = (\mathtt{1Q2J})$. Label the information state corresponding to this history as $s$ depicted by the grey joined circles: $h' = (\mathtt{1Q2K})$. At this information state $s = \{h, h'\}$, it is the fist player's turn ($\tau(s) = 1$) and it includes every history consistent with their information (namely, that they were dealt the jack).

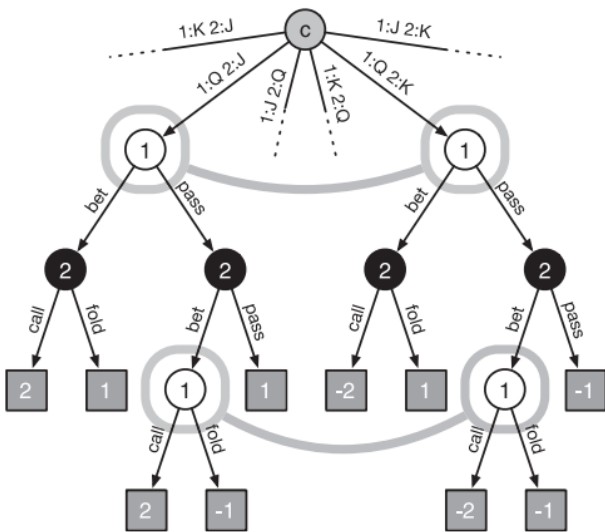

Figure 1: A part of Kuhn poker. Terminal utilities shown for the first player.

The legal actions are now $\mathcal{A}(s) = \{p, b\}$. Suppose the first player chooses $p$ and the second player chooses $b$, then the history is part of $s'$, the second information state shown in the figure. Finally, suppose the first player chooses to bet (call), then the first player would win gaining 2 chips since they have the higher ranking card. Each player $i$'s goal is to compute $\pi_i$ that achieves maximal reward in expectation, where the expectation is taken over all players' policies, even though player $i$ controls only their own policy. Hence, ideally, the player would learn a safe policy that guarantees the best worst-case scenario.

Let $\pi$ denote a joint policy. Define the state-value $v_{\pi,i}(s)$ as the expected (undiscounted) return for player $i$ given that state $s$ is reached and all players follow $\pi$. Let $q_{\pi,i}$ be defined similarly except also conditioned on player $\tau(s)$ taking action $a$ at $s$. Formally, $v_{\pi,i}(s) = \sum_{(h,z) \in \mathcal{Z}(s)} \eta^\pi(h|s)\eta^\pi(h,z)u_i(z)$, where $\mathcal{Z}(s)$ are all terminal histories paired with their prefixes that pass through $s$, $\eta^\pi(h|s) = \frac{\eta^\pi(h)}{\eta^\pi(s)}$, where $\eta^\pi(s) = \sum_{h' \in s} \eta^\pi(h')$, and $\eta^\pi(h,z)$ is the product of probabilities of each action taken by the players' policies along $h$ to $z$. The state-action values $q_{\pi,i}(s,a)$ are defined analogously. Standard value-based RL algorithms estimate these quantities for policy evaluation. Regret minimization in zero-sum games uses a different notion of value, the **counterfactual value**: $v^c_{\pi,i}(s) = \sum_{(h,z) \in \mathcal{Z}(s)} \eta^\pi_{-i}(h)\eta^\pi(h,z)u_i(z)$, where $\eta^\pi_{-i}(h)$ is the product of opponents' policy probabilities along $h$. We also write $\eta^\pi_i(h)$ the product of player $i$'s own probabilities along $h$. Under the standard assumption of perfect recall, we have that for any $h, h' \in s$, $\eta^\pi_i(h) = \eta^\pi_i(h')$. Thus counterfactual values are formally related to the standard values (30): $v_{\pi,i}(s) = \frac{v^c_{\pi,i}(s)}{\beta_{-i}(\pi,s)}$, where $\beta_{-i}(\pi,s) = \sum_{h \in s} \eta^\pi_{-i}(h)$. Also, $q^c_{\pi,i}(s,a)$ is defined similarly except over histories $(ha, z) \in \mathcal{Z}(s)$, where $ha$ is history $h$ concatenated with action $a$.

Counterfactual regret minimization (CFR) is a tabular policy iteration algorithm that has facilitated many advances in Poker AI (35). On each iteration $t$, CFR computes counterfactual values $q^c_{\pi,i}(s,a)$ and $v^c_{\pi,i}(s)$ for each state $s$ and action $a \in \mathcal{A}(s)$ and the

regret of *not* choosing action $a$ (or equivalently the advantage of choosing action $a$ at state $s$, $r^t(s,a) = q^c_{\pi^t,i}(s,a) - v^c_{\pi^t,i}(s)$. CFR tracks the cumulative regrets for each state and action, $R^T(s,a) = \sum_{t=1}^{T} r^t(s,a)$. Define $(x)^+ = \max(0,x)$; regret-matching then updates the policy of each action $a \in \mathcal{A}(s)$ as follows (14):

$$\pi^{T+1}(s,a) = \text{NORMALIZEDRELU}(R^T, s, a) = \begin{cases} \frac{R^{T,+}(s,a)}{\sum_{b\in\mathcal{A}(s)} R^{T,+}(s,b)} & \text{if } \sum_{b\in\mathcal{A}(s)} R^{T,+}(s,b) > 0 \\ \frac{1}{|\mathcal{A}(s)|} & \text{otherwise} \end{cases},$$
(1)

In two-player zero-sum games, the mixture policy $\bar{\pi}^T$ converges to the set of Nash equilibria as $T \to \infty$.

Traditional (off-policy) Monte Carlo CFR (MCCFR) is a generic family of sampling variants (20). In **outcome sampling** MCCFR, a behavior policy $\mu_i$ is used by player $i$, while players $-i$ use $\pi_{-i}$, a trajectory $\rho \sim (\mu_i, \pi_{-i})$ is sampled, and the **sampled counterfactual value** is computed:

$$\tilde{q}^c_{\pi,i}(s,a \mid \rho) = \frac{1}{\eta_i^{(\mu_i,\pi_{-i})}(z)} \eta_i^{(\mu_i,\pi_{-i})}(ha,z) u_i(z),$$
(2)

if $(s,a) \in \rho$, or 0 otherwise. $\tilde{q}^c_{\pi,i}(s,a \mid \rho)$ is an unbiased estim. of $q^c_{\pi,i}(s,a)$ (20, Lemma 1).

However, since these quantities are divided by $\eta_i^{(\mu,\pi_{-i})}(z)$, the product of player $i$'s probabilities, (i) there can be significant variance introduced by sampling, especially in problems involving long sequences of decisions, and (ii) the ranges of the $\tilde{v}^c_i$ can vary wildly (and unboundedly if the exploration policy is insufficiently mixed) over iterations and states, which could make approximating the values in a general way particularly challenging (34). Deep CFR and Double Neural CFR are successful large-scale implementations of CFR with function approximation, and they get around this variance issue by using external sampling or a robust sampling technique, both of which require a perfect game model and enumeration of the tree. This is unfeasible in very large environments or in the RL setting where full trajectories are generated from beginning to the end without having access to a generative model which could be used to generate transitions from any state.

## 2.1 EQUILIBRIA, EXPLOITABILITY, AND NASHCONV

In two-player zero-sum games (and, trivially, single-agent games) a Nash equilibrium policy is optimal because it maximizes a player's worst-case payoff (29). Success in Poker AI, leading to super-human ability, has largely been driven by computing approximate equilibria and playing the strategies against humans.

A Nash equilibrium is a joint policy $\pi^* = (\pi_1^*, \pi_2^*)$ such that no player has incentive to deviate from their respective policy because there is no policy that can achieve higher utility against the opponent's policy. A **best response** for player $i$ is $b_i(\pi_{-i}) = \arg\max_{\pi_i'} u_i(\pi_i', \pi_{-i})$. Finally define player $i$'s **incentive to deviate** (to a best response) as $\delta_i(\pi) = u_i(b_i(\pi_{-i}), \pi_{-i}) - u_i(\pi)$. Then, $\pi$ is a Nash equilibrium if and only if deviating to a best response does not raise a player's utility:

$$\forall i, \delta_i(\pi) = 0.$$

Here, the zero on the right-hand side represents not having any incentive to deviate. However, how about if there is a small amount of incentive? The definition naturally extends to the approximate case where the right-hand size is non-zero. An empirical metric to compute how far an aribtrary policy is to a Nash equilibrium is then the sum over players:

$$\text{NASHCONV}(\pi) = \sum_i \delta_i(\pi) \geq 0.$$

Note that the maximal value for NashConv is twice the utility range (this would occur if each player uses a policy achieving the minimum utility, and there exists a best response which gets the maximum utility). In the poker literature there is a commonly metric called

**exploitability** which computes the average rather than the sum: $\text{EXPLOITABILITY}(\pi) = \frac{\sum_i \delta_i(\pi)}{2}$.

These metrics measure the empirical distance to equilibrium over time leading to an assessment of an algorithm's convergence rate in practice.

## 3    THE ADVANTAGE REGRET-MATCHING ACTOR-CRITIC

ARMAC is a model-free RL algorithm motivated by CFR. Like algorithms in the CFR framework, ARMAC uses a centralized training setup and operates in epochs that correspond to CFR iterations. Like RCFR, ARMAC uses function approximation to generate policies. ARMAC was designed so that as the number of samples per epoch increases and the expressiveness of the function approximator approaches a lookup table, the generated sequence of policies approaches that of CFR. Instead of accumulating cumulative regrets– which is problematic for a neural network– the algorithm learns a conditional advantage estimate $\bar{W}(s, a)$ by regression toward a history-dependent advantage $A(h, a)$, for $h \in s$, and uses it to derive the next set of joint policies that CFR would produce. Indeed we show that $\bar{W}(s, a)$ is an estimate of the cumulative regret $R(s, a)$ up to a multiplicative factor which is a function of the information state $s$ only, and thus cancels out during the regret-matching step. ARMAC is a Monte Carlo algorithm in the same sense as MC-CFR: value estimates are trained from full episodes. It uses off-policy learning for training the value estimates (*i.e.* critics), which we show is sufficient to derive $\bar{W}$. However,

---

**Algorithm 1:** Advantage Regret-Matching Actor-Critic

**input** : initial set of parameters $\boldsymbol{\theta}^0$, num. players $n$
Set initial learning player $i \leftarrow 1$
**for** *epoch* $t \in \{0, 1, 2, \cdots\}$ **do**
    reset $\mathcal{D} \leftarrow \emptyset$
    Let $\pi^t(s) = \text{NORMALIZEDRELU}(\bar{W}_{\boldsymbol{\theta}^t}(s))$
    Let $v_{\boldsymbol{\theta}^t}(h) = \sum_{a \in \mathcal{A}(h)} \pi^t(h, a) q_{\boldsymbol{\theta}^t}(h, a)$
    Let $\mu_i^t$ be a behavior policy for player $i$
    **for** *episode* $k \in \{1, \ldots, K_{act}\}$ **do**
        $i \leftarrow (i + 1) \bmod n$
        Sample $j \sim \text{UNIF}(\{0, 1, \cdots, t - 1\})$
        Sample trajectory $\rho \sim (\mu_i, \pi_{-i}^j)$
        let $d \leftarrow (i, j, \{u_i(\rho)\}_{i \in \mathcal{N}})$
        **for** *history* $h \in \rho$ *where player $i$ acts* **do**
            let $s$ be the state containing $h$
            let $\vec{r} = \{q_{\boldsymbol{\theta}^j}(h, a') - v_{\boldsymbol{\theta}^j}(h)\}_{a' \in \mathcal{A}(s)}$
            let $a$ be the action that was taken in $\rho$
            append $(h, s, a, \vec{r}, \pi^j(s))$ to $d$
        **end**
        add $d$ to $\mathcal{D}$
    **end**
    **for** *learning step* $k \in \{1, \ldots, K_{learn}\}$ **do**
        Sample a random episode/batch $d \sim \text{UNIF}(\mathcal{D})$:
        **for** *history and corresponding state* $(h, s) \in d$ **do**
            Use TB($\lambda$) to train critic $q_{\boldsymbol{\theta}^t}(h, a)$
            If $\tau(s) = i$: train $\bar{W}_{\boldsymbol{\theta}^t}$ to predict $A(h, a)$
            If $\tau(s) \in -i$: train $\bar{\pi}_{\boldsymbol{\theta}^t}$ to predict $\pi^t(s)$
        **end**
    **end**
    Save $\boldsymbol{\theta}^t$ for future retrospective replays; $\boldsymbol{\theta}^{t+1} \leftarrow \boldsymbol{\theta}^t$
**end**

---

contrary to MCCFR, it does *not* use importance sampling. ARMAC is summarized in Algorithm 1.

ARMAC runs over multiple epochs $t$ and produces a joint policies $\pi^{t+1}$ at the end of each epoch. Each epoch starts with an empty data set $\mathcal{D}$ and simulates a variety of joint policies executing multiple training iterations of relevant function approximators. ARMAC trains several estimators which can be either heads on the same neural network, or separate neural networks. The first one estimate the history-action values $q_{\pi^t, i}(h, a) = \sum_{z \in \mathcal{Z}(h, a)} \eta^{\pi^t}(h, z) u_i(z)$. This estimator[2] can be trained on all previous data by using

---

[2]In practice, rather than using $h$ as input to our approximators, we use a concatenation of all players' observations, i.e. an encoding of the *augmented information states* or *action-observation histories* (9; 18). In some games this is sufficient to recover a full history. In others there is hidden state from all players, we can consider any chance event to be delayed until the first observation

any off-policy policy evaluation algorithm from experiences stored in replay memory (we use Tree-Backup($\lambda$) (26)). If trained until zero error, this quantity would produce the same history value estimates as recursive CFR computes in its tree pass. Secondly, the algorithm also trains a state-action network $\bar{W}_i^t(s, a)$ that estimates the expected advantage $A_{\mu^t,i}(h, a) = q_{\mu^t,i}(h, a) - v_{\mu^t,i}(h)$ conditioned on $h \in s$ when following some mixture policy $\mu^t$ (which will be precisely defined in Section B). It happens that $\bar{W}_i^t(s, a)$ is an estimate of the cumulative regret $R^t(s, a)$ multiplied by a (non-negative) function which depends on the information state $s$ only, thus does not impact the policy improvement step by regret-matching (see Lemma 1). Once $\bar{W}_i^t(s, a)$ is trained, the next joint policy $\pi^{t+1}(s, a)$ can be produced by normalizing the positive part as in Eq. 1. After each training epoch the joint policy $\pi^t$ is saved into a past policy reservoir, as it will have to be loaded and played during future epochs. Lastly, an average policy head $\bar{\pi}^t$ is also trained via a classification loss to predict the policy $\pi^{t'}$ over all time steps $t' \leq t$. We explain its use in Section 4.

Using a history-based critic allows ARMAC to avoid using importance weight (IW) based off-policy correction as is the case in MCCFR, but at the cost of higher bias due to inaccuracies that the critic has. Using IW may be especially problematic for long games. For large games the critic will inevitably rely on generalization to produce history-value estimates.

To save memory, reservoir sampling with buffer of size of 1024 was used to prune past policies.

The algorithm also works in a single agent case by treating all opponent reach probabilities as 1. More details and results are given in Appendix in Sections C.1 and D.

Our main theoretical result is that ARMAC learns a function $W^T$ which is a stand-in replacement for the cumulative regrets of CFR, $R^T$. See Appendix B for an analysis of ARMAC's theoretical properties.

A worked out example is given in Appendix in Section A.

## 3.1 ADAPTIVE POLICY SELECTION

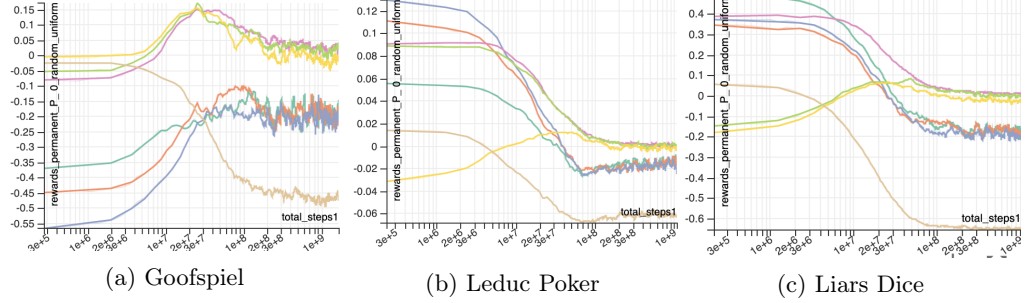

|  (a) Goofspiel | (b) Leduc Poker | (c) Liars Dice |

Figure 2: An average reward per modulations scored against opponent $\bar{\pi}^t$ as a function of time (measured in acting steps). The brown curve is a random uniform policy (i). Cyan, orange and blue are (ii) with $\epsilon \in 0.0, 0.01, 0.05$ respectively. Pink, green and yellow are (iii) with $\epsilon \in 0.0, 0.01, 0.05$.

ARMAC dynamically switches between what policy to use based on estimated returns. For every $t$ there is a pool of candidate policies, all based on the following four policies: (i) random uniform policy. (ii) several policies defined by applying Eq 1 over the current epoch's regret only $(q_{\boldsymbol{\theta}^t}(h, a) - v_{\boldsymbol{\theta}^t}(h))$, with different levels of random uniform exploration: $\epsilon \in 0.0, 0.01, 0.05$ . (iii) several policies defined by the mean regret, $\pi^t$ as stated in Algorithm 1, also with the same level of exploration. (iv) the average policy $\bar{\pi}^t$ trained via classification. ARMAC generates experiences using those policies is to facilitate the problem of exploration and to help produce meaningful data at initial stages of learning before average regrets

---

of its effects by any of the players in the game. Thus, the critics represent an expectation over those hidden outcomes. Since this does not affect the theoretical results, we choose this notation for simplicity. Importantly, ARMAC remains model-free: we never enumerate chance moves explicitly nor evaluate their probabilities which may be complex for many practical applications.

are learnt. Each epoch, the candidate policies are ranked by cumulative return against an opponent playing $\bar{\pi}_{\boldsymbol{\theta}^t}$. The one producing highest rewards is used half of the times. When sub-optimal policies are run for players $-i$, they are not used to train mean regrets for player $i$, but are used to train the critic. Typically, (ii) produces the best policy initially and allows to bootstrap the learning process with the best data (Fig. 2). In later stages of learning, (iii) with the smallest of $\epsilon$ yields better policies and gets consistently picked over other policies. The more complex the game is, the longer it takes for (iii) to take over (ii).

Exploratory policy $\mu_i^T$ is constructed by taking the most recent neural network with 50% probability or otherwise sampling one of the past neural networks uniformly and modulating it by the above described method.

## 3.2 Network architecture

ARMAC can be used with both feed-forward (FF) and recurrent neural networks (RNN) (Fig. 6(a)). For small games where information states can be easily represented, FF networks were used. For larger games, where consuming observations rather than information states is more natural, RNNs were used. More details can be found in Appendix in Section F.

## 4 Empirical Evaluation

For partially-observable multiagent environments, we investigate Imperfect Information (II-) Goofspiel, Liar's Dice, and Leduc Poker and betting-abstracted no-limit Texas Hold'em poker (in Section 4.1). Goofspiel is a bidding card game where players spend bid cards collect points from a deck of point cards. Liar's dice is a 1-die versus 1-die variant of the popular game where players alternate bidding on the dice values. Leduc poker is a two-round poker game with a 6-card deck, fixed bet amounts, and a limit on betting. Longer descriptions of each games can be found in (24). We use OpenSpiel (19) implementations with default parameters for Liar's Dice and Leduc poker, and a 5-card deck and descending points order for II-Goofspiel. To show empirical convergence, we use NashConv, the sum over each player's incentive to deviate to their best response unilaterally (21), which can be interpreted as an empirical distance from Nash equilibrium (reaching Nash at 0).

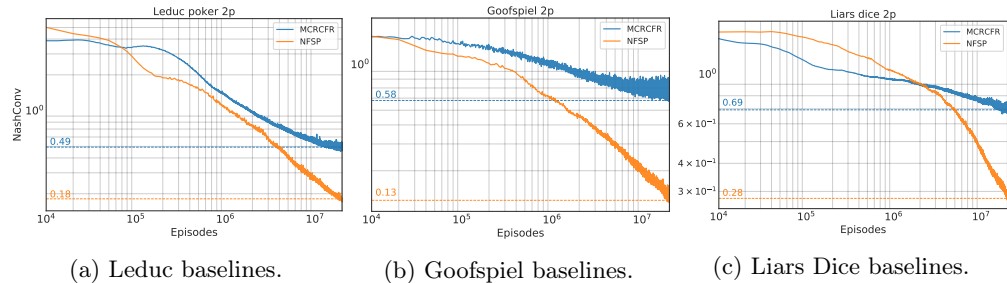

(a) Leduc baselines.     (b) Goofspiel baselines.     (c) Liars Dice baselines.

Figure 3: NFSP and MC-RCFR on the Leduc Poker, II-GoofSpiel with 5 cards and Liars Dice

We compare empirical convergence to approximate Nash equilibria using a model-free sampled form of regression CFR (34) (MC-RCFR). Trajectories are obtained using outcome sampling MCCFR (20), which uses off-policy importance sampling to obtain unbiased estimates of immediate regrets $\hat{r}$, and average strategy updates $\hat{s}$, and individual (learned) state-action baselines (27) to reduce variance. A regressor then predicts $\hat{\bar{R}}$ and a policy is obtained via Eq. 1, and similarly for the average strategy. Each episode, the learning player $i$ plays with an $\epsilon$-on-policy behavior policy (while opponent(s) play on-policy) and adds every datum $(s, \hat{r}, \pi(\hat{\bar{R}}))$ to a data set, $\mathcal{D}$, with a retention rule based on reservoir sampling so it approximates a uniform sample of all the data ever seen. MC-RCFR is related, but not equivalent to, a variant of DeepCFR (5) based on outcome sampling (OS-DeepCFR) (31). Oufar results differ significantly from the OS-DeepCFR results reported in (31), and we discuss differences in assumptions and experimental setup from previous work in Appendix C.

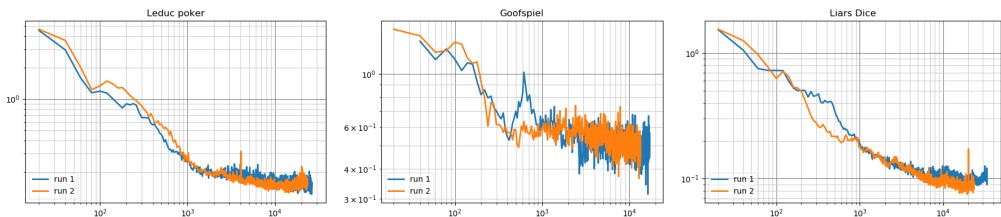

(a) NashConv on Leduc Poker    (b) NashConv on Goofspiel.    (c) NashConv on Liars Dice

Figure 4: ARMAC results on Leduc, II-Goofspiel, and Liar's Dice. The y-axis is NashConv of the average strategy $\bar{\pi}^t$. The x-axis is number of epochs. One epoch consists of 100 learning steps. Each learning step processes 64 trajectories of length 32 sampled from replay memory. The final value reached by the best runs are 0.18 (Leduc), 0.5 (II-Goofspiel), and 0.095 (Liar's Dice).

As with ARMAC, the input is raw bits with no expert features. We use networks with roughly the same number of parameters as the ARMAC experiments: feed-forward with 4 hidden layers of 128 units with concatenated ReLU (28) activations, and train using the Adam optimizer. We provide details of the sweep over hyper-parameters in Appendix C.

Next we compare ARMAC to NFSP (16), which combines fictitious play with deep neural network function approximators. Two data sets, $\mathcal{D}^{RL}$ and $\mathcal{D}^{SL}$, store transitions of sampled experience for reinforcement learning and supervised learning, respectively. $\mathcal{D}^{RL}$ is a sliding window used to train a best response policy to $\bar{\pi}_{-i}$ via DQN. $\mathcal{D}^{SL}$ uses reservoir sampling to train $\bar{\pi}_i$, an average over all past best response policies. During play, each agent mixes between its best response policy and average policy. This stabilizes learning and enables the average policies to converge to an approximate Nash equilibrium. Like ARMAC and MC-RCFR, NFSP does not use any expert features.

Convergence plots for MC-RCFR and NFSP are shown in Figure 3, and for ARMAC in Figure 4. NashConv values of ARMAC are lower (Liar's Dice) and higher (Goofspiel) than NFSP, but significantly lower than MC-RCFR in all cases. MC-RCFR results are consistent with the outcome sampling results in DNCFR (22). Both DNCFR and Deep CFR compensate for this problem by instead using external and robust sampling, which require a forward model. So, next we investigate the performance of ARMAC in a much larger game.

## 4.1 No-Limit Texas Hold'em

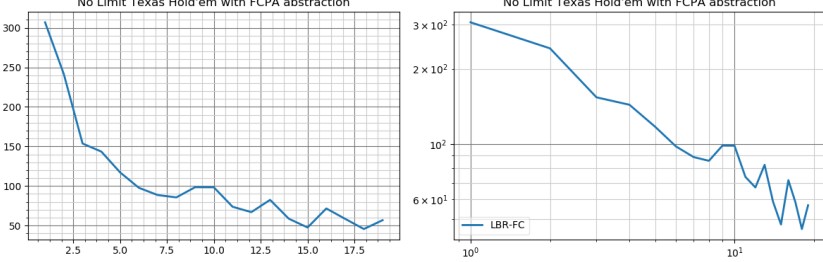

Figure 5: ARMAC results in No-Limit Texas Hold'em trained with FCPA action abstraction evaluated using LBR-FC metric. The y-axis represents the amount LBR-FC wins agains the ARMAC-trained policy. The x-axis indicate days of training. The left graph shows the learning curve in a linear scale, while the right one shows the same curve in a log-log scale.

We ran ARMAC on no-limit Texas Hold'em poker, using the common { Fold, Call, Pot, All-in } (FCPA) action/betting abstraction. This game is orders of magnitude larger than games used above ($\approx 4.42 \cdot 10^{13}$ information states). Action abstraction techniques were used by all of the state-of-the-art Poker AI bots up to 2017. Modern search-based techniques of DeepStack (25) and Libratus (6) still include action abstraction in the search tree.

Computing the NashConv requires traversing the whole game and querying the network at each information state. This becomes infeasible for large games. Instead, we use local best-response (LBR) (23). LBR is an exploiter agent that produces a lower-bound on the exploitability: given some policy $\pi_{-i}$ it does a shallow search using the policy at opponent nodes, and a poker-specific heuristic evaluation at the frontier of the search. LBR found that previous competition-winning abstraction-based Poker bots were far more exploitable than first expected. In our experiments, LBR was limited to the betting abstractions: FCPA, and FC. We used three versions of LBR: LBR-FCPA, which uses all 4 actions within the abstraction, LBR-FC, which uses a more limited action set of { Fold, Call } and LBR-FC12-FCPA34 which uses { Fold, Call } for the first two rounds and FCPA for the rest.

We first computed the average return that an ARMAC-trained policy achieves against uniform random. Over 200000 episodes, the mean value was 516 (chips) $\pm$ 25 (95% c.i.). Similarly, we evaluated the policy against LBR-FCPA; it won 519 $\pm$ 81 (95% c.i.) per episode. Hence, LBR-FCPA was unable to exploit the policy. ARMAC also beat LBR-FC12-FCPA34 by 867 $\pm$ 87 (95% c.i.) . Interestingly, ARMAC learned to beat those two versions of LBR surprisingly quickly. A randomly initialized ARMAC network lost against LBR-FCPA by -704 $\pm$ 191 (95% c.i.) and against LBR-FC12-FCPA34 by -230 $\pm$ 222 (95% c.i.), but was beating both after a mere 1 hour of training by 561 $\pm$ 163 (95% c.i.) and 427 $\pm$ 140 (95% c.i.) respectively ( 3 million acting steps, 11 thousand learning steps).

Counter-intuitively, ARMAC was exploited by LBR-FC which uses a more limited action set. ARMAC scored -46 $\pm$ 26 (95% c.i.) per episode after 18 days of training on a single GPU, 1.3 billion acting steps (rounds), 5 million learning steps, 50000 CFR epochs (Figure 5). To the best of our knowledge, this is the first time LBR has been used to approximate exploitability in any form of no-limit Texas Hold'em among this class of algorithms.

## 5 Conclusion and Future Work

ARMAC was demonstrated to work on both single agent and multi-agent benchmarks. It is brings back ideas from computational game theory to address exploration issues while at the same time being able to handle learning in non-stationary environments. As future work, we intend to apply it to more general classes of multiagent games; ARMAC has the appealing property that it already stores the joint policies and history-based critics, which may be sufficient for convergence one of the classes of extensive-form correlated equilibria (10; 12; 11).

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

# Appendices

## A    WORKED-OUT EXAMPLE

We now show an example of how ARMAC works on the simple game of Kuhn poker, shown in Figure 1.

Suppose ARMAC has already run for $t = 50$ epochs, so 50 networks have been saved, and the exploring player is the first player $i = 1$. The first player uses an exploratory behavior policy $\mu_i^t$ as described above. The second player uses network $j = 17$ sampled from $\text{UNIF}(\{0, 1, \cdots, 49\})$. For this episode, chance samples 1Q2K. This happens with probability one sixth, so $\eta_{-i}(h) = \frac{1}{6}$ (chance is always seen as an opponent with a fixes policy) whereas player 1 has not taken any actions so $\eta_i(h) = 1$. Along this episode $\rho$, the first player samples actions according to $\mu_i$ and the second player according to $\pi_{-i}^{17}$. Suppose then player 1 samples bet and player 2 samples bet (call) leading to $u_1(\rho) = -2$ for player 1 and $u_2(\rho) = 2$ for player 2. There are two histories traversed, call them $h$ and $h'$ respectively. For each one, the regret vectors $\vec{r}$ are determined by the critics $q_{\boldsymbol{\theta}^j}(h, a') - v_{\boldsymbol{\theta}^j}(h)$, where $a'$ is one the two legal actions. Trajectory $\rho$ is added to the buffer $\mathcal{D}$ and many similar episodes take place.

Finally, in the learning phase: ARMAC uses all the data collected to train the critics using standard $\ell_2$ regression losses on the TD error defined by $\text{TB}(\lambda)$; all the data can be used because TB is off-policy, allowing the exploratory behavior $\mu_i^{50}$. Suppose examples from the first trajectory $\rho$ are sampled: only data from the first player (history $h$) are used to train $\bar{W}_{\boldsymbol{\theta}^t}$ on the advantage $A(h, a)$ using standard regression loss; this is because to asymptotically approach CFR only the exploring player can train regrets leading to a scaling constant that is a function only of the information state (for more detail, see Appendix B). Finally, only the second player's actions are used to train the average network $\bar{\pi}$ using a classification loss, as the second player in $\rho$ was playing according to CFR's average policy across 50 epochs (due to sampling $j$ uniformly and then playing $\pi_{-i}^j$ without exploration).

## B    THEORETICAL PROPERTIES

Each epoch $t$ estimates $q_{\pi^t, i}(h, a) = \sum_{z \in \mathcal{Z}(h, a)} \eta^{\pi^t}(h, z) u_i(z)$ and value $v_{\pi^t, i}(h) = \sum_a \pi^t(h, a) q_{\pi^t, i}(h, a)$ for the current policies $(\pi^t)$. Let us write the advantages $A_{\pi^t, i}(h, a) = q_{\pi^t, i}(h, a) - v_{\pi^t, i}(h)$. Notice that we learn functions of the history $h$ and not state $s$.

At epoch $T$, in order to deduce the next policy, $\pi^{T+1}$, CFR applies regret-matching using the cumulative counterfactual regret $R_i^T(s, a)$. As already discussed, directly estimating $R_i^T$ using sampling suffers from high variance due to the inverse probability $\eta_i^{(\mu, \pi_{-i})}(z)$ in (2). Instead, ARMAC trains a network $\bar{W}_i^T(s, a)$ that estimates a conditional advantage along trajectories generated in the following way: For player $i$ we select a behavior policy $\mu_i^T$ providing a good state-space coverage, *e.g.* a mixture of past policies $(\pi_i^t)_{t \leq T}$, with some added exploration (Section 3.1 provides more details). For the other players $-i$, for every trajectory, we choose one of the previous opponent policies $\pi_{-i}^j$ played at some epoch $j$ chosen uniformly at random from $\{1, 2, \cdots T\}$. Thus at epoch $T$, several trajectories $\rho^j$ are generated by following policy $(\mu_i^T, \pi_{-i}^j)$, where $j \sim \mathcal{U}(\{1, 2, \cdots T\})$.

Then at each step $(h, a)$ along these trajectory $\rho^j$, the neural network estimate $\bar{W}_i^T(s, a)$ (where $s \ni h$) is trained to predict the advantage $A_{\pi^j, i}(h, a)$ using the empirical $\ell_2$ loss: $\hat{\mathcal{L}} = \left[ \bar{W}_i^T(s, a) - A_{\pi^j, i}(h, a) \right]^2$. Thus the corresponding average loss is

$$\mathcal{L} = \frac{1}{T} \sum_{j=1}^{T} \mathbb{E}_{\rho^j \sim (\mu_i^T, \pi_{-i}^j)} [\hat{\mathcal{L}}] = \frac{1}{T} \sum_{j=1}^{T} \sum_{s \in \mathcal{S}_i} \sum_{h \in s} \eta^{(\mu_i^T, \pi_{-i}^j)}(h) \mu_i^T(s, a) \left[ \bar{W}_i^T(s, a) - A_{\pi^j, i}(h, a) \right]^2.$$

If the network has sufficient capacity, it will minimize this average loss, and $\bar{W}_i^T(s, a)$ will converge (when the number of trajectories goes to $\infty$) in each state-action pair $(s, a)$, such

that the reach probability $\frac{1}{T}\sum_t \eta^{(\mu_i^T, \pi_{-i}^t)}(s)\mu_i^T(s,a) > 0$, to the conditional expectation

$$W_i^T(s,a) = \sum_{h\in s} \frac{\frac{1}{T}\sum_{j=1}^T \eta^{(\mu_i^T, \pi_{-i}^j)}(h)A_{\pi^k,i}(h,a)}{\frac{1}{T}\sum_{j=1}^T \eta^{(\mu_i^T,\pi_{-i}^t)}(s)} \underbrace{=}_{\text{perfect recall}} \sum_{h\in s} \frac{\frac{1}{T}\sum_{j=1}^T \eta_{-i}^{\pi^j}(h)A_{\pi^k,i}(h,a)}{\frac{1}{T}\sum_{j=1}^T \eta_{-i}^{\pi^t}(s)} \tag{3}$$

Notice that $W_i^T$ does not depend on the exploratory policy $\mu_i^T$ for player $i$ chosen in round $T$. After several trajectories $\rho^j$ our network $\bar{W}_i^T$ provides us with a good approximation of the $W_i^T$ values and we use it in a regret matching update to define the next policy, $\pi_i^{T+1}(s) = \text{NORMALIZEDReLU}(\bar{W}_i^T)$, *i.e.* Equation 1. Lemma 1 shows that if $\bar{W}_i^T(s,a)$ is sufficiently close to the $W_i^T(s,a)$ values, then this is equivalent to CFR, i.e., doing regret-matching using the cumulative counterfactual regret $R^T$.

**Lemma 1.** *The policy defined by* $\text{NORMALIZEDReLU}(W_i^T)$ *is the same as the one produced by CFR when regret matching is employed as the information-state learner:*

$$\pi_i^{T+1}(s,a) = \frac{R_i^{T,+}(s,a)}{\sum_b R_i^{T,+}(s,b)} = \frac{W_i^{T,+}(s,a)}{\sum_b W_i^{T,+}(s,b)}. \tag{4}$$

*Proof.* First, let us notice that

$$W_i^T(s,a) = \sum_{h\in s} \frac{\sum_{t=1}^T \eta^{\pi^t}(h)}{\sum_{t=1}^T \eta^{\pi^t}(s)} A_{\pi^t,i}(h,a), \tag{5}$$

$$= \sum_{h\in s} \frac{\sum_{t=1}^T \eta_{-i}^{\pi^t}(h)}{\sum_{t=1}^T \eta_{-i}^{\pi^t}(s)} A_{\pi^t,i}(h,a) \tag{6}$$

$$= \frac{1}{w^T(s)} \sum_{t=1}^T \sum_{h\in s} \eta_{-i}^{\pi^t}(h)A_{\pi^t,i}(h,a), \tag{7}$$

where we used the perfect recall assumption in the first derivation, and we define $w^T(s) = \sum_t \eta_{-i}^{\pi^t}(s)$. Notice that $w^T(s)$ depends on the state only (and not on $h$). Now the cumulative regret is:

$$R_i^T(s,a) = \sum_{t=1}^K q_{\pi^t,i}^c(s,a) - v_{\pi^t,i}^c(s)$$

$$= \sum_{t=1}^T \eta_{-i}^{\pi^t}(s)\left(q_{\pi^t,i}(s,a) - v_{\pi^k,i}(s)\right)$$

$$= \sum_{t=1}^T \eta_{-i}^{\pi^t}(s) \sum_{h\in s} \frac{\eta_{-i}^{\pi^t}(h)}{\eta_{-i}^{\pi^t}(s)} \left(q_{\pi^t,i}(h,a) - v_{\pi^t,i}(h)\right)$$

$$= \sum_{t=1}^T \sum_{h\in s} \eta_{-i}^{\pi^t}(h)A_{\pi^t,i}(h,a)$$

$$= w^T(s)W_i^T(s,a).$$

Finally, noticing that regret matching is not impacted by multiplying the cumulative regret by a positive function of the state, we deduce

$$\frac{R_i^{T,+}(s,a)}{\sum_b R_i^{T,+}(s,b)} = \frac{\left(w^T(s)W_i^T(s,a)\right)^+}{\sum_b \left(w^T(s)W_i^T(s,b)\right)^+} = \frac{W_i^{T,+}(s,a)}{\sum_b W_i^{T,+}(s,b)}.$$

$\square$

The $\bar{W}^T(s,a)$ estimates the expected advantages $\frac{1}{T}\sum_{j=1}^T A_{\pi^j}(h,a)$ conditioned on $h\in s$. Thus ARMAC does not suffer from the variance of estimating the cumulative regret $R^T(s,a)$, and in the case of infinite capacity, from any $(s,a)$, the estimate $\bar{W}^T(s,a)$ is unbiased as soon as the $(s,a)$ has been sampled at least once:

**Lemma 2.** *Consider the case of a tabular representation and define the estimate $\hat{W}_i^T(s, a)$ as the minimizer (over $W$) of the empirical loss defined over $N$ trajectories*

$$\hat{\mathcal{L}}_{(s,a)}(W) = \frac{1}{N} \sum_{n=1}^{N} \left[W - A_{\pi^{j_n},i}(h, a)\right]^2 \mathbb{I}\{(h, a) \in \rho^{j_n} \text{ and } h \in s\},$$

*where $\rho^{j_n}$ is the n-th trajectory generated by the policy $(\mu_i^T, \pi_{-i}^{j_n})$ where $j_n \sim \mathcal{U}(\{1, \ldots, T\})$. Define $N(s, a) = \sum_{n=1}^{N} \mathbb{I}\{(h, a) \in \rho^{j_n} \text{ and } h \in s\}$ to be the number of trajectories going through $(s, a)$. Then $\hat{W}_i^T(s, a)$ is an unbiased estimate of $W_i^T(s, a)$ conditioned on $(s, a)$ being traversed at least once:*

$$\mathbb{E}\left[\hat{W}_i^T(s, a) | N(s, a) > 0\right] = W_i^T(s, a).$$

*Proof.* The empirical loss being quadratic, under the event $\{N(s, a) > 0\}$, its minimum is well defined and reached for

$$\hat{W}_i^T(s, a) = \frac{1}{N(s, a)} \sum_{n=1}^{N(s,a)} A_{\pi^{j_n},i}(h_n, a),$$

where $h_n \in s$ is the history of the $n$-th trajectory traversing $s$. Let us use simplified notations and write $A_n = A_{\pi^{j_n},i}(h, a)\mathbb{I}\{(h, a) \in \rho^{j_n} \text{ and } h \in s\}$ and $b_n = \mathbb{I}\{(h, a) \in \rho^{j_n} \text{ and } h \in s\}$. Thus

$$
\begin{aligned}
\mathbb{E}\left[\hat{W}_i^T(s, a)\mathbb{I}\left\{\sum_{m=1}^{N} b_m > 0\right\}\right] &= \mathbb{E}\left[\frac{\sum_{n=1}^{N} A_n\mathbb{I}\{\sum_{m=1}^{N} b_m > 0\}}{\sum_{m=1}^{N} b_m}\right] \\
&= \sum_{n=1}^{N} \mathbb{E}\left[\mathbb{E}\left[\frac{A_n\mathbb{I}\{\sum_{m=1}^{N} b_m > 0\}}{\sum_{m=1}^{N} b_m} \Big| \sum_{m=1}^{N} b_m\right]\right] \\
&= \sum_{n=1}^{N} \mathbb{E}\left[\mathbb{E}\left[A_n \Big| \sum_{m=1}^{N} b_m\right]\frac{\mathbb{I}\{\sum_{m=1}^{N} b_m > 0\}}{\sum_{m=1}^{N} b_m}\right]
\end{aligned}
$$

Now, $\mathbb{E}\left[A_n | \sum_{m=1}^{N} b_m\right] = \mathbb{E}[A_n|b_n]\mathbb{E}\left[b_n | \sum_{m=1}^{N} b_m\right]$ since given $b_n$, $A_n$ is independent of $\sum_{m=1}^{N} b_m$. Thus

$$
\begin{aligned}
\mathbb{E}\left[\hat{W}_i^T(s, a)\mathbb{I}\left\{\sum_{m=1}^{N} b_m > 0\right\}\right] &= \sum_{n=1}^{N} \mathbb{E}[A_n|b_n]\mathbb{E}\left[\frac{\mathbb{E}\left[b_n | \sum_{m=1}^{N} b_m\right]\mathbb{I}\{\sum_{m=1}^{N} b_m > 0\}}{\sum_{m=1}^{N} b_m}\right] \\
&= \sum_{n=1}^{N} \mathbb{E}[A_n|b_n]\mathbb{E}\left[\frac{b_n\mathbb{I}\{\sum_{m=1}^{N} b_m > 0\}}{\sum_{m=1}^{N} b_m}\right]
\end{aligned}
$$

Since $\sum_{n=1}^{N} \mathbb{E}\left[\frac{b_n\mathbb{I}\{\sum_{m=1}^{N} b_m > 0\}}{\sum_{m=1}^{N} b_m}\right] = \mathbb{E}\left[\frac{\sum_{n=1}^{N} b_n}{\sum_{m=1}^{N} b_m}\mathbb{I}\{\sum_{m=1}^{N} b_m > 0\}\right] = \mathbb{P}\left(\sum_{m=1}^{N} b_m > 0\right)$, by a symmetry argument we deduce $\mathbb{E}\left[\frac{b_n\mathbb{I}\{\sum_{m=1}^{N} b_m > 0\}}{\sum_{m=1}^{N} b_m}\right] = \frac{1}{N}\mathbb{P}\left(\sum_{m=1}^{N} b_m > 0\right)$ for each $n$. Thus

$$
\begin{aligned}
\mathbb{E}\left[\hat{W}_i^T(s, a)\Big|N(s, a) > 0\right] &= \mathbb{E}\left[\hat{W}_i^T(s, a)\Big|\sum_{m=1}^{N} b_m > 0\right] \\
&= \frac{\mathbb{E}\left[\hat{W}_i^T(s, a)\mathbb{I}\left\{\sum_{m=1}^{N} b_m > 0\right\}\right]}{\mathbb{P}\left(\sum_{m=1}^{N} b_m > 0\right)} \\
&= \frac{1}{N}\sum_{n=1}^{N} \mathbb{E}[A_n|b_n] = \mathbb{E}[A_1|b_1]
\end{aligned}
$$

which is the expectation of the advantage $A_{\pi^j,i}(h, a)$ conditioned on the trajectory $\rho^j$ going through $h \in s$, i.e. $W_i^T(s, a)$ as defined in (3). $\qquad\square$

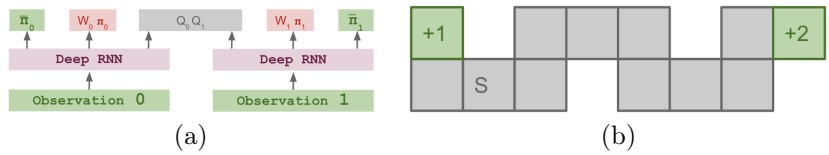

Figure 6: The (a) Multi-headed network architecture, and (b) Exploration example.

## C    Baseline Details and Hyperparameters

For MC-RCFR, we sweep over all combinations of the exploration parameter, using a (learned) state-action baseline (27), and learning rate $(\epsilon, b, \alpha) \in \{0.25, 0.5, 0.6, 0.7, 0.9, 0.95, 1.0\} \times \{\text{TRUE}, \text{FALSE}\} \times \{0.0001, 0.00005, 0.00001\}$, where each combination is averaged over five seeds. We found that higher exploration values worked consistently better, which matches the motivation of the robust sampling technique (corresponding to $\epsilon = 1$) presented in (22) as it leads to reduced variance since part of the correction term is constant for all histories in an information state. The baseline helped significantly in the larger game with more variable-length episodes.

For NFSP, we keep a set of hyperparameters fixed, in line with (21) and (16): anticipatory parameter $\eta = 0.1$, $\epsilon$-greedy decay duration $20M$ steps, reservoir buffer capacity $2M$ entries, replay buffer capacity $200k$ entries, while sweeping over a combination of the following hyperparameters: $\epsilon$-greedy starting value $\{0.06, 0.24\}$, RL learning rate $0.1, 0.01, 0.001$, SL learning rate $\{0.01, 0.001, 0.005\}$, DQN target network update period of $\{1000, 19200\}$ steps (the later is equivalent to 300 network-parameter updates). Each combination was averaged over three seeds. Agents were trained with the ADAM optimizer, using MSE loss for DQN and one gradient update step using mini-batch size 128, every 64 steps in the game.

Finally, note that there are at least four difference in the results, experimental setup, and assumptions between MC-RCFR and OS-DeepCFR reported in (31):

1. (31) uses domain expert input features which do not generalize outside of poker. The neural network architecture we use is a basic MLP with raw input representations, whereas (31) uses a far larger network. Our empirical results on benchmark games compare the convergence properties of knowledge-free algorithms across domains.

2. The amount of training per iteration is an order of magnitude larger in OS-DeepCFR than our training. In (31), every 346 iterations, the Q-network is trained using 1000 minibatches of 512 samples (512000 examples), whereas every 346 iterations we train 346 batches of 128 samples, 44288 examples.

3. MC-RCFR uses standard outcome sampling rather than Linear CFR (7).

4. MC-RCFR's strategy is approximated by predicting the OS's average strategy increment rather than sampling from a buffer of previous models.

Our NFSP also does not use any extra enhancements.

### C.1    Single-Agent Environments

Despite ARMAC being based on commonly-used multiagent algorithms, it has properties that may be desirable in the single-agent setting. First, similar to policy gradient algorithms in the common "short corridor example" (32, Example 13.1), stochastic policies are representable by definition, since they are normalized positive mean regrets over the actions. This could have a practical effect that entropy bonuses typically have in policy gradient methods, but rather than simply adding arbitrary entropy, the relative regret over the set of past policies is taken into account.

Second, a retrospective agent uses a form of *directed exploration* of different exploration policies (2). Here, this is achieved by the simulation $(\mu_i^T, \pi_{-i}^t)$, which could be desirable whenever there is overlapping structure in successive tasks. $\mu_i^T$ here is an exploratory

policy, which consists of a mixture of all past policies (plus random uniform) played further modulated with different amounts of random uniform exploration (more details are given in Section 3.1). Consider a gridworld illustrated in Fig. 6(b). Green squares illustrate positions where the agent $i$ gets a reward and the game terminates. Most of RL algorithms would find the reward of $+1$ first as it is the closest to the origin $S$. Once this reward is found, a policy would quickly learn to approach it, and finding reward $+2$ would be problematic. ARMAC, in the meantime, would keep re-running old policies, some of which would pre-date finding reward $+1$, and thus would have a reasonable chance of finding $+2$ by random exploration. This behaviour may also be useful if instead of terminating the game, reaching one of those two rewards would start next levels, both of which would have to be explored.

These properties are not necessarily specific to ARMAC. For example, POLITEX (another retrospective policy improvement algorithm (1)) has similar properties by keeping its past approximators intact. Like POLITEX, we show an initial investigation of ARMAC in Atari in Appendix D. Average strategy sampling MCCFR (13) also uses exploration policies that are a mixture of previous policies and uniform random to improve performance over external and outcome sampling variants. However, this exact sampling method cannot be used directly in ARMAC as it requires a model of the game.

## D  INITIAL INVESTIGATION OF ARMAC IN THE ATARI LEARNING ENVIRONMENT

While performance on Atari is not the main contribution, it should be treated as a health check of the algorithm. Unlike previously tested multiplayer games, many Atari games have a long term credit assignment problem. Some of them, like Montezuma's Revenge, are well-known hard exploration problems. It is interesting to see that ARMAC was able to consistently score 2500 points on Montezuma's Revenge despite not using any auxiliary rewards, demonstrations, or distributional RL as critic. We hypothesize that regret matching may be advantageous for exploration, as it provides naturally stochastic policies which stay stochastic until regrets for other actions becomes negative. We also tested the algorithm on Breakout, as it is a fine control problem. We are not claiming that out results on Atari are state of art - they should be interpreted as a basic sanity check showing that ARMAC could in principle work in this domain.

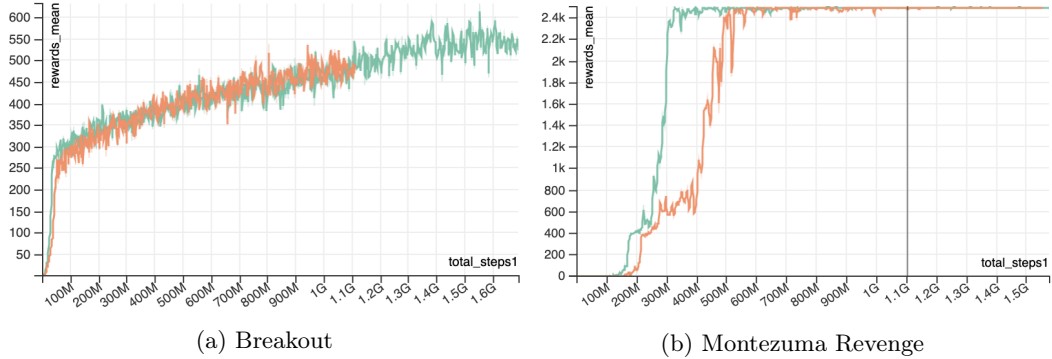

(a) Breakout                    (b) Montezuma Revenge

Figure 7: Performance on Breakout (left) and Montezuma Revenge (right). Results are shown for two seeds.

## E  TRAINING

Training is done by processing a batch of 64 of trajectories of length 32 at a time. In order to implement a full recall, all unfinished episodes will be continued on the next training iteration by propagating recurrent network states forward. Each time when one episode finishes at a particular batch entry, a new one is sampled and started to be unrolled from the beginning.

Adam optimized with $\beta_1 = 0.0$ and $\beta_2 = 0.999$ was used for optimization. Hyperparameter selection was done by trying only two learning rates: $5 \cdot 10^{-5}$ and $2 \cdot 10^{-4}$. The results reported use $5 \cdot 10^{-5}$ in all games, including Atari.

## F  NEURAL NETWORK ARCHITECTURE

The following recurrent neural network was used for no-limit Texas Hold'em experiments. Two separate recurrent networks with shared parameters were used, consuming observations of each player respectively. Each of those networks consisted of a single linear layer mapping input representation to a vector of size 256. This was followed by a double rectified linear unit, producing a representation of size 512 then followed by LSTM with 256 hidden units. This produced an information state representation for each player $a_0$ and $a_1$.

Define architecture $B(x)$, which will be reused several times. It consumes one of the information state representations produced by the previously mentioned RNN: $h_1 = Linear(128)(x)$, $h_2 = DoubleReLU(h_1)$, $h_3 = h1 + Linear(128)(h2)$, $B(a) = DoubleReLU(h_3)$.

The immediate regret head is formed by applying $B(s)$ on the information state representation followed by a single linear layer of the size of the number of actions in the game. The same is done for an average regret head and mean policy head. All those $B(s)$ do not share weights between themselves, but share weights with respective heads for another player.

The global critic $q(h)$ is defined in the following way. $n_A = Linear(128)$, $n_B = Linear(128)$, $a_0 = n_A(s_0) + n_B(s_1)$, $a_0 = n_B(s_0) + n_A(s_1)$, $h_1 = Concat(a_0, a_1)$, $h_2 = B(h_1)$ and finally $q_0(s_1, s_2)$ and $q_1(s_1, s_2)$ are evaluated by a two linear layers on top of $h_2$. $B(x)$ shares architecture but does not share parameters with the ones used previously.

