# OpenReview forum: "The Advantage Regret-Matching Actor-Critic"
_ICLR.cc/2021/Conference — Reject_

### Official Review · AnonReviewer2 · 2020-10-18
**Requires improved presentation and bit more experimentation**

**Rating:** 6
**Confidence:** 3

**Review:**

This paper considers the problem of counterfactual regret minimization and proposes an algorithm that does not use the importance sampling procedure. The claim is that this helps in reducing the variance usually introduced by the IS procedure. They propose a new algorithm that uses the previously used policies as a buffer and replays those policies to learn a new policy. The algorithm is also claimed to be highly scalable for games with large state-action pairs.


My overall assessment of this paper is that, the problem considered is an important one and a well-implemented/well-written paper would definitely be a good paper that is significantly impactful for this community. However, my personal opinion is that this paper is not yet there. First, the paper is not well-written for someone who is not familiar with the exact prior literature on CFR using deep learning. In particular, the paper is not organized well, some of the important definitions are omitted and in general the key points aren't sufficiently highlighted. Also I find the experiments to be a bit lacking. In particular, one of the key claims of the paper is that it eliminates the variance introduced by the IS procedure, yet there are no experiments to substantiate this (more below). Also the executed experiments are not explained well. My opinion is that the section of "theoretical properties" is totally unhelpful for the main section and can be relegated to the appendix; it does not add value to the understanding of this algorithm. Additionally, I have several comments below that can help with both the writing, the weaker experiments and overall improving this paper to a state that will make it more impactful/easier to understand. My current assessment of this paper is that without a major revision of the writing and execution, this paper is not in a state to be accepted.


(1) A self-sufficient description of the problem statement. The paper does describe the problem, but it takes a few readings for a new reader not familiar with the exact line of work to get the setup and the contributions of this paper. Moreover, as an arrangement it is spread over introduction and notations. Finally, the paper doesn't explicitly state what the goal of a CFR algorithm is. Is it to minimize the total average counterfactual regret?

(2) Along those lines, I think the paper would be very informative if the algorithm was evaluated and instantiated on a very simple two-player game with  known Nash equilibria (even simpler than Atari and Montezuma's revenge). In particular, there are many such games known in theory (e.g., the prisoner's dillema) and you can pick one of them and instantiate/evaluate this algorithm to help the reader understand the notations precisely. My opinion is that the notations is intertwined with informal descriptions and it is often hard to parse what the authors are meaning to say. Having such a unified simple game will make this process easier on the reader. This also ensures that the implementation and the algorithm is itself correct.

(3) Some definitions in the empirical section are not defined formally. In particular, the quantity NashConv is not defined in this paper and relegated to a related work. Only the informal description is given. This makes the reading really hard. For instance, we know that it should be close to 0, but how close? For instance, how do I interpret a value of 0.5? What is the range of NashConv? These questions could be easily answered if a precise definition of NashConv was included in this paper.

(4) Issues with experiments: First, in the convergence plot for ARMAC in Figure 3, it would be good to plot the line for what the NashConv is (like in Figure 2). Second, the main claim of this paper is that the variance introduced by importance sampling is not present (because this algorithm does not invoke the IS procedure). However, it is surprising that the empirical evaluation section does not include a convincing experiment to drive this point. In particular, it would be good to have a comparison of the variance of the three algorithms and show that the variance introduced by the IS procedure is indeed a meaningful problem for the final outcome (i.e., variance in the convergence value to the NashConv) and that the proposed method indeed reduces it? If the variance introduced by the IS procedure does not lead to a meaningful variance on NashConv, please motivate better why the variance introduced by IS is a problem?

(5) Some suggestions on plots: I find Figure 1 to have particularly difficult to read color schemes. First, the caption states a Pink line and it took me a while to figure which was Pink (I think it is the violet line which is pink). The same problem with Brown. Overall, the 7 colors used are pretty close and I would suggest either trying to use other markers to distinguish (such as a text on the line, different line type etc) or try and use contrasting color schemes. For instance, some of these lines may be indistinguishable for a person with color blindness. Likewise, the titles of the plots are not meaningful/interpretable. In figure 1, both the x and y axis seem like titles that were used internally by the authors. I do not understand what they mean. In the x-axis title, I don't see what that 1 represents and I quite frankly do not understand what the y-axis is trying to say. Likewise, among the three plots, across Figures 1, 2, 3 if the order of the three environments remained consistent, that would help the readability. This is also important, because I still do not understand how to interpret Figure 1. I think 0 implies that the algorithm is good. But if its negative does it mean that the algorithm is better while if its positive it is worse? Please explain Figure 1 better.

(6) Finally there are many typos throughout the paper. Here is a non-exhaustive list.
- page 3: heads on the same neural -> I don't understand what this statement is saying and seems grammatically incorrect as stated.
- Page 3: The first one estimate -> The first one estimates
- In caption of Figure 1, the word modulations is used. This word is not defined anywhere else in the paper. Please either define this or re-use a word that has been introduced.
- Page 6: ARMAC generates experiences using those... -> this sentence is grammatically incorrect. Please fix.
- Page 8: Conclusions. "It is brings back" -> grammar check this sentence
- Page 8: Conclusions: "for convergence one of the classes" -> grammar check

---

> ### Author Response · Authors · 2020-11-24
> **Response to AnonReviewer2**
>
> Thank you for the thorough review. We have applied significant modifications based on your review, which we describe below.
>
> Textual Modifications:
>
> - New or modified text since the previous copy is highlighted in blue. This will be changed to black in the final copy.
> - We have added a diagram of a simple game to the Background and an example playing of that game to describe all the terms
> - We have added a step-by-step walkthrough example of ARMAC in Appendix using this simple game.
> - We moved the entire Theoretical Properties section to the appendix.
> - We have added text to the intro to clarify CFR and a new paragraph clearly stating the problem statement.
>
> Point 1:
>
> We have now added a sentence at the end of the first paragraph of the intro to explain the goal of CFR (but yes, also: each player minimizes their cumulative counterfactual regret; in self-play, this leads to approximate equilibria on average).
>
> We have added a problem statement paragraph at the end of the intro which is self-contained, clear problem the paper addresses.
>
> Point 2:
>
> The benchmark games used in the first part of Section 4 are much smaller than Atari games but non-trivial since they require playing stochastic policies at equilibrium, standard RL algorithms for MDPs are not applicable. These are games that have commonly been used across this literature and are small enough that we can compute the exact distance to Nash equilibrium (this is the metric NashConv). Prisoner’s dilemma is a non-zero sum one-shot game. In this paper, we examine zero-sum extensive games, which are sequential (extended in time), hence the motivation of using RL.
>
> We have added a step-by-step example walk-through of the algorithm using Kuhn poker, a very small extensive-form game that can be explained via a diagram. We do not include results on Kuhn poker in our results because it is too small and any principled RL algorithm solves it (finds a Nash equilibrium with high precision) almost immediately. We placed the worked out example in the appendix to verbosity, but we may as well move it into the main text upon request.
>
> Point 3:
>
> We have added Subsection 2.1 that defines Nash equilibria and the empirical metrics used to compute the convergence rates of algorithms in practice (NashConv, exploitability).
>
> Point 4:
>
> ARMAC accomplishes two things as far as variance reductions is concerned. Firstly, it defined a new quantity W that has a similar scale for all information states. This is unlike what vanilla counterfactual regrets are like, that become negligible small for states far away from the root node of the game tree due to low reach probabilities, that are equal to the product of all action probabilities from the start of the game up to that state. While this makes no difference  in a tabular setting, this is very problematic for neural networks that are not good at expressing quantities of very different scale. DeepCFR, for instance, also solves this problem, but by renormalizing counterfactual regrets by a different constant.
>
> In order to measure empirical variance, at first we have to define what we want to measure. What matters most, is the variance of gradients that a given optimizer has to deal with, and gradient values are proportional to L1 losses when L2 loss is minimized (this is because a 2x is a derivative of x^2). Also, while absolute scales of losses or gradients do not matter (as they are absorbed by the optimizer), only variability of their magnitudes matters, as the learning rate has to be able to handle the worst possible jumps. Thus, we defined the following quantity to measure: signal to noise ratio. Say we sample the batch of size B and we train the mean regret estimator for each valid action within that information state. We have A actions in total. For each batch entry and each action, we evaluate an appropriate L1 loss (gradient magnitude estimate) of the regret head. Then we calculate the following statistics across all batch entries, all actions and many training iterations: mean(L1) and mean(L1^2). We define var(L1) = mean(L1^2) - mean(L1)^2. We define sdev(L1) = sqrt(var(L1)). And finally, we define signal to noise ratio as S = mean(L1) / sdev(L1). The lower this ratio is, the higher the effective sample size within a given batch is, and the more efficient learning process will be.
>
> We evaluated this quantity for on Leduc Poker and Liars Dice . ARMAC got  0.40 on Leduc Poker and 0.41 on liars dice.
>
> We then computed the same metric for the regret network in MC-RCFR, obtaining 0.35 for Leduc and 0.18 for Liar’s Dice. Note that the drop compared to ARMAC is considerably larger on Liar’s dice, which is a larger game (~24 times larger than Leduc), with more variable length episodes (3-14 vs. 7-12).

---

### Official Review · AnonReviewer4 · 2020-10-24
**Good combination with gaming theory and actor-critic, but still having several concerns**

**Rating:** 6
**Confidence:** 3

**Review:**

In this paper, the authors adopt the idea from gaming theory to reinforcement learning and propose a new algorithm that uses the previous policy to update the current training without using importance sampling. Experiments show that the proposed algorithm cannot only work on the single-player setting but also work on the multi-agent (zero-sum) problems. However, I have the following concerns about the algorithms:
1) To train the critic, how would the sample complexity be? Like vanilla Actor-Critic algorithms, can it be replaced by doing a one-step TD(0) update on the critic to improve the sample efficiency?
2) Since the original CFR method is solving the multi-agent zero-sum algorithm, it would be interesting why this extension could solve the single-player problem?

Also, it would make the contribution of this work more clear if the author can compare this exploration method with other exploration methods, such as $\epsilon$-greedy or UCB. To me, the algorithm uses sample trajectory $\rho \sim (\mu_i, \pi_{-i}^j)$. If we make all $\mu$ be random policy, is this exploration similar to $\epsilon$-greedy to some extent?

Considering all contributions and concerns mentioned above, I will suggest a borderline accept for this paper. I might change my score after the author's response and discussion.

---

> ### Author Response · Authors · 2020-11-24
> **Our response to AnonReviewer4**
>
> Thank you very much for your helpful review.
>
> Addressing point 1.
> From the perspective of critic training, the problem is a pure RL and sample complexity will ultimately depend on the algorithm used. In this paper we used TB(lambda) (tree-Backup by Precup et al) algorithm, but any other off-policy policy evaluation algorithm can be used instead. We do not think that our choice of the algorithm was optimal but we chose it only for algorithmic simplicity.
>
> Addressing point 2.
> CFR can also solve single player settings by substituting other players (player -i) reach probability with 1. A single player game is just a special case of a two player game where the second player does not make any move. CFR can certainly solve such games, as long as they have a finite state space. For the same reason ARMAC is also capable of solving single agent problems and we tested on a few ATARI domains as a sanity check.
>
> The main contribution behind ARMAC is the realization that by storing policy pools (as opposed to a replay of experiences) and training a critic network one can accurately evaluate all necessary reach probabilities that are necessary to derive a neural version of CFR without using any importance weights.
>
> Also, ARMAC is compatible with many exploration methods that can be applied in a single agent case as well. In depth exploration of different exploration strategies was not the main focus of the paper. By exploring exploratory strategies too deeply we may lose the main focus. The key thing we wanted to show in that section is that a bandit chooses to use mean regret policies freely over recent regret policies.

---

### Official Review · AnonReviewer3 · 2020-11-02
**ARMAC is model-free algorithm and improved from neural based CFR.**

**Rating:** 6
**Confidence:** 5

**Review:**

Review:
This paper proposes a general model-free RL method for no-regret learning based on a repeated reconsideration of past behavior. The ARMAC algorithm using the off-policy policy evaluation algorithm TreeBackup to estimate value function and use regret matching to get the next joint policy.

This paper idea is origin from DCFR, DNCFR, single CFR.  But those are model-based algorithms. ARMAC is model-free algorithms and it can be used in a more border environment.  If the author compares this paper to the DREAM algorithm(Deep Regret minimization with Advantage baselines and Model-free learning) to state ARMAC advantage, I will update my score.

---

> ### Author Response · Authors · 2020-11-24
> **Out response to AnonReviewer3**
>
> Thank you very much for your helpful review.
>
> Reference to DREAM paper: https://arxiv.org/pdf/2006.10410.pdf (DREAM: Deep Regret minimization with Advantage
> baselines and Model-free learning)
>
> We have not compared our work with DREAM experimentally as DREAM is not an asymptotically consistent implementation of CFR: it does not preserve the convergence guarantees of MCCFR. DREAM would give biased results even in tabular settings. The key problem in their paper is the absence of all necessary adaptations when transitioning from DeepCFR (that uses external sampling), to DREAM (that uses outcome sampling). When an algorithm is executing external sampling (i.e. expanding all branches in player i decision nodes) and putting observations into reservoir memory, it is in expectation equivalent to sampling actions uniformly at player i’s decision point. This key here is that sampling policy for player i actions does not change across training epochs. However, when outcome sampling is used, sampling policy of the player i changes with training epochs and this requires much extra work to compensate for.
>
> DREAM algorithm does not store past policies neither for player i nor for player -i. Instead, the algorithm stores experiences in reservoir memory B (described in Section 4.5 for the case of DeepCFR and mentioned in section 5.2 in the case of DREAM implementation) where both sampling player policies are correlated. And, unlike in the case of DeepCFR using external sampling, empirical frequencies using DREAMS outcome sampling of a given information state visitations will no longer be independent of players i’s policy run at time t and will be proportional to the total reach of both players at that epoch. Section 5.2 of DREAM’s paper contains the following statement:
>
> “Furthermore, because from agent i’s perspective histories are sampled based on their information, the expectation for data added to B_d i in infostate s_i is proportional to r^t_i (s_i , a_i). “
>
> The statement is correct only when a single epoch’s values are averaged over. However, once average regrets are calculated across more than one epoch by using data within the reservoir replay, each epoch's contribution towards average regrets of a given information state will be weighted by player’s i policy pi^t_{i}(s, a) as well as -i.
>
> Very similar problem applies when an average policy is calculated in section 5.3 . CFR calculates an average policy weighted by player’s i reach probability only. Such averaging only works when the policy of an opponent is not changing in time. Such calculation was not problematic for DeepCFR, as external sampling emulates a time independent uniform sampling policy. However, when outcome sampling is used, both policies within the replay memory get correlated. Technically, DREAM paper averages policy weighting it by the total reach and not only player’s i reach and thus is incompatible with CFR. This can be fixed by using appropriate importance weights, but this would lead to a lot of variance and has not been done in DREAM’s paper.
>
> In our work we spent a lot of effort making sure that all those calculations are done correctly and that our algorithm would produce correct reach weightings for every quality we evaluate. The necessity of making both player policies independent in each training epoch, and thus allowing appropriate quantities to be eliminated was one of the reasons we chose to store all past policies and repeatedly resample them from game play. This can not be trivially done by only using a large (reservoir) replay memory and is one of the main contributions of ARMAC.
>
> Finally, note that we have run ARMAC on FCPA no-limit Hold’em, showing for the first time local best-response (LBR) bounds on exploitability, whereas DREAM showed results on a hand-constructed poker subgame. No other neural RL algorithm has shown decreasing exploitability of any form over training time in this game. Only DeepStack has reported similar metrics, and it used search and even more domain knowledge than DREAM + Deep CFR. We consider our poker result to be a significant achievement, and a testament to the scalability that we designed ARMAC around.

---

> > ### Author Response · Authors · 2021-01-18
> > **DREAM is a sound implementation of CFR**
> >
> > We spoke to the original authors and realized that there was a misunderstanding on our part. We retract our claim of DREAM not being sound. Apologies to the authors and reviewers.

---

### Decision · Program_Chairs · 2021-01-07
**Final Decision**

**Decision:**

Reject

**Comment:**

This paper introduces a new algorithm to solve game, more or less similar (in the general idea, yet differences are interesting) than CFR. The concept is to sample from past policies to generate trajectories and update sequentially (via regret matching).

The three reviewers gave rather lukewarm reviews, with possible suggestions of improvements (that were more or less declined by the authors for those proposed by Rev3 and Rev4; the added material focuses more on the clarity of the text than on the content itself).

I have also read the paper, and find it quite difficult to assess. At the end, it is not clear to a reader whether ARMAC is the new state of the art, or just a "variant" of CFR that will be soon forgotten. The performances do not seem astonishing (at least against NSFP) and even though DREAM might not be satisfactory to the authors (EDIT POST DISCUSSION: actually, DREAM is a valid competitor and must be included in the comparative study), it would have been nice to provide some comparison. Maybe the issue is the writing of the paper that could and should be improved so that it is clearer what are the different building blocks of ARMAC (and their respective importance).

If ARMAC is the new state of the art, then I am sure the authors will be able to clearly illustrate it in a forthcoming revision (maybe with more experiments, as suggested by Rev2). Unfortunately, for the moment, I do not think this paper is mature enough for ICLR.